# The Role of ROR1 in Chemoresistance and EMT in Endometrial Cancer Cells

**DOI:** 10.3390/medicina59050994

**Published:** 2023-05-21

**Authors:** Kyung-Jun Lee, Nam-Hyeok Kim, Hyeong Su Kim, Youngmi Kim, Jae-Jun Lee, Jung Han Kim, Hye-Yon Cho, Soo Young Jeong, Sung Taek Park

**Affiliations:** 1Institute of New Frontier Research Team, Hallym University, Chuncheon 24252, Republic of Korea; rudwns0222@naver.com (K.-J.L.);; 2Division of Hemato-Oncology, Department of Internal Medicine, Kangnam Sacred-Heart Hospital, Hallym University Medical Center, Hallym University College of Medicine, Seoul 07441, Republic of Korea; 3Department of Anesthesiology and Pain Medicine, Chuncheon Sacred-Heart Hospital, Hallym University Medical Center, Hallym University College of Medicine, Chuncheon 24253, Republic of Korea; 4Department of Obstetrics and Gynecology, Dongtan Sacred-Heart Hospital, Hallym University Medical Center, Hallym University College of Medicine, Hwaseong 18450, Republic of Korea; 5Department of Obstetrics and Gynecology, Kangnam Sacred-Heart Hospital, Hallym University Medical Center, Hallym University College of Medicine, Seoul 07441, Republic of Korea

**Keywords:** endometrial cancer, ROR1, receptor tyrosine kinase-like orphan receptor, epithelial-mesenchymal transition (EMT), chemoresistance

## Abstract

*Background and Objectives*: Receptor tyrosine kinase-like orphan receptor type 1 (ROR1) plays a critical role in embryogenesis and is overexpressed in many malignant cells. These characteristics allow ROR1 to be a potential new target for cancer treatment. The aim of this study was to investigate the role of ROR1 through in vitro experiments in endometrial cancer cell lines. *Materials and Methods*: ROR1 expression was identified in endometrial cancer cell lines using Western blot and RT-qPCR. The effects of ROR1 on cell proliferation, invasion, migration, and epithelial-mesenchymal transition (EMT) markers were analyzed in two endometrial cancer cell lines (HEC-1 and SNU-539) using either ROR1 silencing or overexpression. Additionally, chemoresistance was examined by identifying MDR1 expression and IC50 level of paclitaxel. *Results*: The ROR1 protein and mRNA were highly expressed in SNU-539 and HEC-1 cells. High ROR1 expression resulted in a significant increase in cell proliferation, migration, and invasion. It also resulted in a change of EMT markers expression, a decrease in E-cadherin expression, and an increase in Snail expression. Moreover, cells with ROR1 overexpression had a higher IC50 of paclitaxel and significantly increased MDR1 expression. *Conclusions*: These in vitro experiments showed that ROR1 is responsible for EMT and chemoresistance in endometrial cancer cell lines. Targeting ROR1 can inhibit cancer metastasis and may be a potential treatment method for patients with endometrial cancer who exhibit chemoresistance.

## 1. Introduction

Uterine cancer is the second most common gynecologic cancer in women worldwide, and its incidence has been increasing [1]. Endometrial cancer (EC), which originates from endometrial glandular epithelial cells, is the most common type of uterine cancer and includes endometrioid (83%), serous (4–6%), clear cell (1–2%), and other minor types. Additionally, carcinosarcoma, also called malignant mixed mullerian tumor (MMMT), accounts for approximately 4% of all uterine cancers [2].

Early-stage EC patients show favorable survival outcomes, with a 5-year survival rate of 80–95% with appropriate treatment. Initial treatment of early-stage EC is surgical treatment, total hysterectomy, and bilateral salpingo-oophorectomy, and adjuvant treatment is usually radiotherapy. However, patients with advanced-stage or recurrent disease have a poor prognosis [3]. Moreover, owing to their aggressive characteristics, some histologic types of EC, especially carcinosarcoma, do not respond to standard treatment, surgery, or radiotherapy. Approximately 50% of patients with EC who had poor outcomes (cancer recurrence or death) had non-endometrioid cancer types [4]. However, histologic types have not completely reflected the prognosis of EC. According to recent ESGO-ESTRO-ESP guidelines, newly-presented diagnostic algorithms using three immunohistochemical markers (p53, MSH6, and PMS2) and one molecular test (mutation analysis of the exonuclease domain of POLE) have been encouraged in EC for predicting prognosis based on TCGA molecular-based classification and for determining new treatments [5]. It means that there is still no obvious predictive marker of treatment response in EC. Thus, individualized approaches targeting membrane proteins or genetic mutations in EC are emerging for cancer treatment [6].

The Wnt signaling pathway is a developmental pathway in embryogenesis, regulating cancer cell polarity, differentiation, and migration during carcinogenesis [7,8,9]. There are two distinct arms in this pathway; the β-Catenin dependent (canonical) and the β-Catenin independent (noncanonical) arm [10]. Through mutations in β-Catenin or APC, the dependent pathway has been related to the development of endometrial, breast, colon, and gastric cancer [11,12,13]. β-Catenin mutations are seen in approximately 26% of endometrioid EC but not in serous ECs [7]. The independent pathway can be activated by binding from the Wnt5a ligand to receptor tyrosine kinase-like orphan receptor type 1 (ROR1) receptor and mediates enhanced tumor cell growth [14].

ROR1, a transmembrane glycoprotein, is a key protein in embryogenesis and is only expressed in certain normal tissues. However, several studies have shown that it is overexpressed in malignant cells, such as leukemia, breast cancer, prostate cancer, and ovarian cancer [15,16,17,18,19,20,21]. Therefore, this surface antigen has been proposed as a potential novel target for cancer treatment. Patients with advanced-stage or recurrent status of EC or p53 mutation in molecular classification may have a poor response to standard treatment. Hence, this study aimed to investigate whether ROR1 has therapeutic potential in EC using several in vitro experiments.

## 2. Materials and Methods

### 2.1. Cell Lines and Reagents

In this study, two uterine cancer cell lines, EC (Ishikawa, HEC-1) and carcinosarcoma (SNU-539, SNU-685), were obtained from the Korean Cell Line Bank (Seoul, Republic of Korea). The genetic information, database name, and accession number are shown in Appendix A. Ishikawa cells were cultured in DMEM (Welgene, Gyeongsan-si, Republic of Korea), and HEC-1 cells were cultured in McCoy’s 5A medium (Welgene, Gyeongsan-si, Republic of Korea). SNU-539 and SNU-685 cells were cultured in RPMI1640 medium (Welgene, Gyeongsan-si, Republic of Korea). All cell culture media contained 10% fetal bovine serum (FBS) (Invitrogen, Waltham, MA, USA) and 1% penicillin and were incubated in humidified air at 37°C with 5% CO_2_. The transfection reagent jetPRIME^®^ was purchased from PolyPlus-transfection (Illkirch, France), and the transduction reagent polybrene was purchased from Merck Millipore (Burlington, MA, USA). The antibodies used were E-cadherin (Cell Signaling Technology, Danvers, MA, USA), Snail (Cell Signaling Technology, Danvers, MA, USA), ROR1 (Cell Signaling Technology, Danvers, MA, USA), and actin (Cell Signaling Technology, Danvers, MA, USA). The CCK-8 assay was performed using the Viability Assay Kit (Medifab, Geumcheon-gu, Republic of Korea). Paclitaxel, which is used as an anticancer agent, was purchased from Sigma-Aldrich (Burlington, MA, USA).

### 2.2. Establishment of Taxol-Resistant EC Cell Line (Ishikawa^Taxol^)

The paclitaxel-resistant endometrial adenocarcinoma cell line, Ishikawa^Taxol^, was established utilizing a gradient concentration increment method. The Ishikawa cells were given paclitaxel, and the dead cells were removed. The remaining viable cells were then labeled as drug-resistant. Higher paclitaxel doses were subsequently added to the culture medium for the cells that were still viable. Ishikawa^Taxol^ cells were able to survive in a final culture medium containing 0.05 μg/mL of paclitaxel with a steady rise in paclitaxel. Then, Ishikawa^Taxol^ cells were cultured in a medium with stable resistance in the absence of any drug for 2–3 weeks before experimentation. To confirm chemoresistance, the IC50 values of Ishikawa and Ishikawa^Taxol^ cells were compared after 48 h, and paclitaxel was administered at different concentrations to confirm whether there was a statistically significant difference in cell line construction.

### 2.3. Generation of ROR1 Overexpression Stable Cell Lines

The Lentiviruses with ROR1 overexpression or a control vector were purchased from Vectorbuilder (Chicago, IL, USA). In 6-well plates, HEC-1 cell cultures were performed. The medium containing lentiviruses and polybrene (8 µg /mL) was given to the cells at a multiplicity of infection (MOI) of 10 after they had reached 60% confluence and were thoroughly mixed. To increase the effectiveness of infection, polybrene was used. The infection medium was replaced with fresh McCoy’s 5A medium after 24 h, and 48 h after infection, selection pressure (1 μg/mL puromycin) was applied. Puromycin-resistant cell colonies were chosen after 3 to 4 weeks and then established larger in the selection medium to produce HEC-1 clones produced from single cells.

### 2.4. siRNA Preparation and Inhibition of ROR1 Expression

ROR1 inhibition was induced using a siRNA. ROR1 and control siRNAs were designed and synthesized by Bioneer (Daejeon, Republic of Korea). The siRNA duplexes used were as follows: ROR1 siRNA, 5′-CAG GAU ACU CAG AUG AGUdTdT-3′ (sense) and 5′-UAC UCA UCU GAG UAU CCU GdTdT-3′ (antisense). Cells were plated in 6-well plates at a 50% confluence per well and maintained in an RPMI-1640 medium supplemented with 10% FBS at 37 °C with 5% CO_2_. siRNAs (50 nM) were transfected with jetPRIME^®^ (Polyplus-transfection, Illkirch, France) according to the manufacturer’s recommendations. Transfection and silencing efficiencies were monitored using western blotting.

### 2.5. Quantitative Real-Time PCR

Easy-Blue (Invitrogen, Waltham, MA, USA) was used to collect total RNA from cells, and a reverse transcription reagent kit was used to generate first-strand cDNA of the RNAs. Real-time PCR was performed using the SYBR Green PCR kit (TaKaRa, Kusatsu, Japan) in a Rotor-Gene Q (Qiagen, Hilden, Germany). The following primers were utilized: ROR1 (forward), 5′-AACAGACACAGAGTGTGACCT-3′ and ROR1 (reverse), 5′-TAAGGTGTGAACTCTGGGGAG-3′; actin (forward), 5′-AAACTGGAACGGTGAAGG-3′ and actin (reverse), 5′-TGCAATCAAAGTCCTCGG-3′. RNA expression was calculated using the 2-∆∆Ct method and normalized to actin expression. Each assay was performed three times.

### 2.6. Western Blotting

To extract total protein lysates, radio-immunoprecipitation assay lysis buffer containing protease inhibitors was used. The cleared lysates were quantified using a bicinchoninic acid protein assay kit. Proteins were separated using sodium dodecyl sulfate-polyacrylamides gel electrophoresis and electrophoretically transferred to polyvinylidene difluoride membranes (Bio-Rad, Hercules, CA, USA). Membranes were blocked with 5% skim milk for 1 h and incubated with anti-E-cadherin (1:5000 dilution), anti-Snail (1:2000 dilution), anti-ROR1 (1:2000 dilution), or anti-actin (1:10,000 dilution) antibodies at 4 °C overnight. The membranes were received with the corresponding horseradish peroxidase-linked secondary antibody the following day. Membranes were cleaned three times for 10 min each, after which the blots were identified using an enhanced Western Pico ECL Kit (LPS solution, Burlington, MA, USA). Subsequently, they were exposed to ImageQuant LAS 500 (Cytiva, Marlborough, MA, USA). Image J software (version 1.53k) was used to measure the band area density change.

### 2.7. Cell Viability (CCK Assay)

The relative viability of cells was identified by WST-8 [2-(2-methoxy-4-nitrophenyl)-3-(4-nitrophenyl)-5-(2,4-disulfophenyl)-2H-tetrazolium, monosodium salt] (CCK-8) assay. First, cancer cells were counted, and 1500–2000 cells per well were cultivated in a 96-well cell culture plate for 24 h. Then, the CCK-8 solution (Medium: CCK-8 solution 10:1) was supplemented to each well, 10 μL of CCK-8 solution was added to each well following reagent treatment, and the cells were then left to incubate for 2 h. An optical microplate reader was used to detect absorbance at 450 nm. All assays were conducted in triplicate.

### 2.8. Drug Treatment

Cells (3–5 × 10^3^ cells/well) were seeded in 96-well plates and cultured for 24 h. Paclitaxel was diluted to a range of concentrations (10^−4^–10 μM) in a cell culture medium added with 2% FBS and then supplemented to the wells. The cells were incubated for 48 h after 10 μL of WST-8 [2-(2-methoxy-4-nitrophenyl)-3-(4-nitrophenyl)-5-(2,4-disulfophenyl)-2H-tetrazolium, monosodium salt] was added to each well. After 2 h of incubation at 37 °C, the cells were examined to determine their relative viability by measuring the optical density at 450 nm and comparing viability to the paclitaxel-free control.

### 2.9. Wound Healing Assay

The migratory capacity of EC was measured using a wound healing assay. Transfected or transduced cells (5 × 10^5^ cells) were plated in 6-well plates and incubated in a cell culture medium with 10% FBS at 37 °C until reaching 100% confluence. With scraping with a 200 µL tip, the cells were injured. They were then washed three times in a serum-free medium and incubated in a regular medium for 48 h. The wounds were observed at the start time and 48 h incubation. The wound area at each time point was subtracted from the wound area at the 0 h time point to determine the cell migratory distance. All assays were conducted in triplicate.

### 2.10. Invasion Assay

A transwell chamber (8 um pore size; SPL, Pocheon-si, Republic of Korea) and Matrigel invasion (Corning, Glendale, AZ, USA) were used in cell migration and invasion assay, respectively. Cells in the serum-free media were positioned in the upper chamber and covered with 2–10 ug/mL Matrigel 48 h after transfection or transduction. The medium containing 10% FBS was added to the lower chamber. Cells that had not migrated or remained in the upper part after 48 h of incubation were removed. Cells that had migrated or invaded were fixed in 4% paraformaldehyde, stained with 0.1% crystal violet, and counted under a microscope. All assays were conducted three times.

### 2.11. Immunofluorence

In 12-well plates, cells seeding on glass coverslips were fixed in a 4% formaldehyde solution and permeabilized with 0.5% Triton X-100/PBS. After blocking with 5% horse serum PBS for 30 min at room temperature, cells were incubated with 2% horse serum primary antibodies (anti-E-cadherin, 1:500 dilutions; anti-Snail, 1:200 dilutions; or anti-ROR1, 1:200 dilutions) at 4 °C overnight. They were then incubated with fluorescent dye-conjugated secondary antibodies for 1 h. With DAPI staining, an inverted fluorescence microscope captured the stained images.

### 2.12. Statistical Analysis

All experiments were performed a minimum of three times, and data are presented as the mean ± Standard Error of the Mean (SEM). Prism (version 8.0) was used for all the statistical analyses. Two-group comparisons were performed using Student’s *t*-test. Multiple group comparisons were performed using one-way analysis of variance (ANOVA). All tests were two-sided. Statistical significance was set at *p* < 0.05.

## 3. Results

### 3.1. ROR1 Is Responsible for Expression of Epithelial-Mesenchymal Transition (EMT) Factors and Chemoresistance in EC Cell Lines

ROR1 expression was identified in Ishikawa, HEC-1, SNU-539, and SNU-685 EC cell lines using western blotting and RT-qPCR. The HEC-1 cell line, a model of Type II endometrial adenocarcinoma, and the SNU-539 cell line, a model of carcinosarcoma, exhibited higher expression of ROR1 protein and mRNA. Additionally, the expression of ROR1 and EMT markers showed a similar trend in the HEC-1 and SNU-539 cell lines with higher ROR1 expression, lower E-cadherin expression, and higher Snail expression (Figure 1A). Additionally, the IC50 values of paclitaxel in each cell line varied and showed different chemoresistance. Concerning ROR1 expression, higher levels were related to Taxol resistance (Figure 1B). Immunofluorescence analysis revealed the localization of ROR1 and EMT markers in the cells (Figure 1C). Original western blot replicates and densitometry readings/intensity ratio of each band are shown in Appendix A, and the experimental values are in Table 1.

### 3.2. ROR1 Expression Regulates the Invasion, Migration, and Tumorigenic Potential In Vitro

The SNU-539 cell line, a carcinosarcoma cell line derived from a uterine malignant mixed Müllerian tumor, exhibits a high expression of ROR1. ROR1 silencing was performed using siRNA. ROR1 silencing in SNU-539 cells significantly decreased cell migration and invasion. In the migration assay, the mean difference between control and ROR1-silenced groups at 48 h was 49.57% (95% confidence interval (CI) 33.60–65.54, *p* < 0.0001) (Figure 2A). In the invasion assay, the mean difference between the two groups was 149.0 ± 6.221 (*p* < 0.0001) (Figure 2B). Cell viability was also analyzed after incubation for 24, 48, 72, or 96 h in the CCK assay. As shown in Figure 2C, ROR1 silencing significantly decreased SNU-539 cell proliferation. The mean difference of absorbance in 96 h was 1.194 (95% CI 1.038–1.351, *p* < 0.0001). In the colony-forming assay, ROR1 silencing resulted in fewer and smaller colonies than in the control cells. The mean difference between the two groups was 36.67 ± 2.625 (*p* = 0.0002) (Figure 2D).

HEC-1 cells were engineered to overexpress ROR1 as a model of Type II endometrial adenocarcinoma. Overexpression of ROR1 significantly increased cell migration and invasion. In the migration assay, the mean difference between control and ROR1-overexpressed groups at 48 h was 35.43% (95% CI 19.50–51.37, *p* = 0.0006) (Figure 2E). In the invasion assay, the mean difference between the two groups was 548.4 ± 30.06 (*p* < 0.0001) (Figure 2F). Moreover, the proliferation of HEC-1 cells was measured by the CCK assay, and ROR1 overexpressed cells showed significant proliferation (Figure 2G). The mean difference of absorbance in 96 h was 0.6036 (95% CI 0.4159–0.7912, *p* < 0.0001). In the colony assay, ROR1 overexpression showed in more and larger colonies than in the control cells. The mean difference between the two groups was 393.7 ± 22.60 (*p* < 0.0001) (Figure 2H).

### 3.3. ROR1 Has an Effect on the Sensitivity to the AntiCancer Drug

We investigated the changes in EMT markers, including E-cadherin and Snail, in the SNU-539 cell line with ROR1 silencing and in the HEC-1 cell line with ROR1 overexpression. E-cadherin overexpression and reduced Snail expression were shown in ROR1-silenced SNU-539 cells (Figure 3A and Table 2). In the HEC-1 cells, overexpression of ROR1 was significantly related to lower E-cadherin expression and greater Snail1 expression (Figure 3B and Table 3). These data revealed that ROR1 expression could be significantly associated with EMT and promote metastatic potential in EC.

To examine the resistance to the anticancer drug, we identified the IC50 level of paclitaxel and the drug resistance marker, MDR1, in two cell lines. In the SNU-539 cell line, ROR1-silenced cells had a lower IC50 of Taxol (58.18 vs. 6221 nM); however, there was no significant difference in MDR1 expression levels (Figure 3C). In the HEC-1 cell line, ROR1-overexpressed cells had higher IC50 of Taxol (175.3 vs. 26.14 nM) and significantly increased MDR1 expression (Figure 3D). Original western blot replicates and densitometry readings/intensity ratio of each band are shown in Appendix A.

### 3.4. Paclitaxel-Resistant Uterine Cancer Cells Show Higher Levels of ROR1 Expression

The paclitaxel-resistant cell line (Ishikawa^Taxol^ cells) had stronger cancer proliferation and metastasis abilities than Ishikawa cells. In migration and invasion assays, Ishikawa^Taxol^ cells showed greater tumor metastatic ability than Ishikawa cells. Additionally, the Ishikawa^Taxol^ group showed increased cell proliferation in the CCK assay, resulting in more numerous and larger colonies in the colony assay (Appendix A).

ROR1, EMT markers, and MDR1 expression were identified using western blotting and RT-qPCR in the Ishikawa and Ishikawa^Taxol^ cell lines (Figure 4A and Table 4). Ishikawa^Taxol^ cell lines exhibited higher expression of ROR1, lower E-cadherin expression, and higher Snail expression. High MDR1 expression was identified in the Ishikawa^Taxol^ cell line, and the IC50 of Ishikawa^Taxol^ cells was significantly higher than that of Ishikawa cells (4611 vs. 767.3 nM) (Figure 4B). Original western blot replicates and densitometry readings/intensity ratio of each band are shown in Appendix A, and the experimental values are in Appendix A.

## 4. Discussion

In this study, we investigated the effects of ROR1 on cell proliferation and metastasis, the expression between ROR1 and EMT, and the effects of ROR1 on chemoresistance in EC cells. First, we analyzed ROR1 and related EMT markers, E-cadherin and Snail, in EC cell lines. We then examined the effects of ROR1 silencing and overexpression in two EC cell lines. Finally, we investigated the relationship between ROR1 expression and chemoresistance.

ROR1 is a type-I orphan receptor tyrosine kinase-like surface protein that plays an important role in embryogenesis. Many studies have shown that ROR1 was highly expressed in several malignant cells, and the actual function of ROR1 in cancer was in search [15,16,17,18,19,20,21]. It was first identified as an oncoembryonic gene in hematological malignancies [22]. After that, the importance of ROR1 function in chronic lymphocytic leukemia (CLL) cells was introduced by Fukada et al. [23]. ROR1 expression showed a significant prognostic factor in patients with CLL, and several studies identified aberrant expression of ROR1 in other hematologic malignancies, diffuse large B cell lymphoma, follicular lymphoma, and marginal zone lymphoma [24]. Moreover, ROR1 has been shown to cancer cell survival and proliferation in various solid tumors. High expression of ROR1 was associated with EMT and tumor metastasis in breast cancer cells [16,19] and showed poor prognosis in lung adenocarcinoma [10]. Additionally, high ROR1 expression in ovarian cancer cells showed stem cell-like gene-expression profiles [21]. There have also been investigated that high expression of ROR1 in other solid tumors, including colorectal cancer, gastric cancer, melanoma, and pancreatic cancer, was associated with poor prognosis [15,16,17,18,19,20,21].

Several studies have investigated the expression of ROR1 in EC, and high expression of ROR1 has been identified in blood samples and paraffin-embedded tissues from patients with EC [25,26,27]. However, the role of ROR1 in EC remains to be elucidated. Therefore, we examined the effects of ROR1 on cell proliferation, colony formation, and metastasis using plasmids to silence or overexpress ROR1 in human EC cell lines. High ROR1 expression was significantly associated with the proliferative and metastatic abilities of EC cells. These results suggest that ROR1 is involved in EC pathogenesis.

EMT is a biological process that enables tumor cells to become invasive and migrate to self-renewal, progression, and metastasis [28]. This is related to the hypoxic environment, cytokines, signaling pathways, including the Wnt pathway, and several transcription factors [29]. ROR1 can contribute to the β-catenin-independent Wnt pathway to activate cancer cell growth by binding to the Wnt5a ligand [7,30]. These results suggest that ROR1 is associated with EMT. Cui et al. revealed that ROR1 silencing in breast cancer cell lines enhanced the expression of E-cadherin, epithelial cytokeratins, and tight-junction proteins and impaired their migration/invasion capacity [16]. One of the EMT markers, Snail, a zinc finger transcription factor, usually upregulates the expression of E-cadherin [31]; therefore, a combination of E-cadherin and Snail can be hallmarking features of EMT [29]. In this study, we found that expression of ROR1 was significantly associated with EMT markers, loss of E-cadherin, and activation of Snail. These results suggested that treatments targeting ROR1 can potentially inhibit cancer metastasis.

Several studies have reported that high ROR1 expression is significantly associated with cancer survival [19,21,25,26,32,33,34,35,36,37,38]. Similar results were observed for EC. Henry et al. reported that high ROR1 group had poor overall survival (hazard ratio (HR) 3.74, 95% confidence interval (CI) 1.54–9.07, *p* = 0.004) [25], and Liu et al. also revealed the same results (HR for progression-free survival 2.45, 95% CI = 1.21–4.97, *p* = 0.01; HR for overall survival 2.48, 95% CI = 0.99–6.18, *p* = 0.05) [26]. Previous studies have shown that ROR1 is abundant in chemoresistant stem cells. Fultang et al. showed that ROR1 regulates chemoresistance in chemoresistant breast cancer cell lines by modulating ABCB1, a drug efflux pump [39]. Additionally, upregulation of ROR1 has been identified in chemoresistant ovarian cancer cells [40,41]. In the present study, we found that high ROR1 expression was significantly associated with chemoresistance, with the expression of MDR1 protein, and a comparison of the IC50 of paclitaxel. Therefore, ROR1 expression may be involved in poor survival and chemoresistance.

In summary, ROR1 is involved in EC pathogenesis, particularly the EMT process, which inhibits cancer metastasis and is highly related to chemoresistance. New cancer treatments have emerged that target ROR1, such as monoclonal antibodies (mAbs), anti-body-drug conjugates (ADCs), or chimeric antigen receptor (CAR)-T cells. UC-961 (cirmtuzumab) is the first humanized mAb targeting ROR1 in CLL [42], and clinical trials of single or combination therapy with other agents are ongoing (NCT03088878, NCT02860676, NCT04501939, and NCT02776917). VLS-101 (combination of mAb and monomethyl auristatin E) and NBE-002 (combination of mAb and PNU-159682) are ADCs [43] that have been evaluated in clinical trials (NCT04504916, NCT03833180, and NCT04441099). NVG-111 is a bispecific antibody that targets ROR1 and CD3 [44], and a phase I/II trial is ongoing (NCT04763083). ROR1-CAR-specific autologous T-lymphocytes have been developed, and a phase I trial is ongoing for leukemia/lymphoma, non-small cell lung carcinoma, and breast cancer (NCT02706392). Clinical trials are mainly conducted for hematologic malignancies, breast cancer, or non-small cell lung carcinoma. Based on previous studies and the results of this study, ROR1-targeted treatment can also be considered in EC, especially for subtypes with poor prognosis or chemoresistant characteristics. However, our findings need to be verified with additional clinical specimens derived from EC patients.

## 5. Conclusions

In summary, our results provide evidence that ROR1 increases cell proliferation and cancer cell metastatic ability. The results also show that overexpression of ROR1 in taxol-resistant EC cells led to the drug-resistance. Our findings suggest that ROR1 can be a potential therapeutic target for overcoming the drug resistance seen in EC.

## Figures and Tables

**Figure 1 medicina-59-00994-f001:**
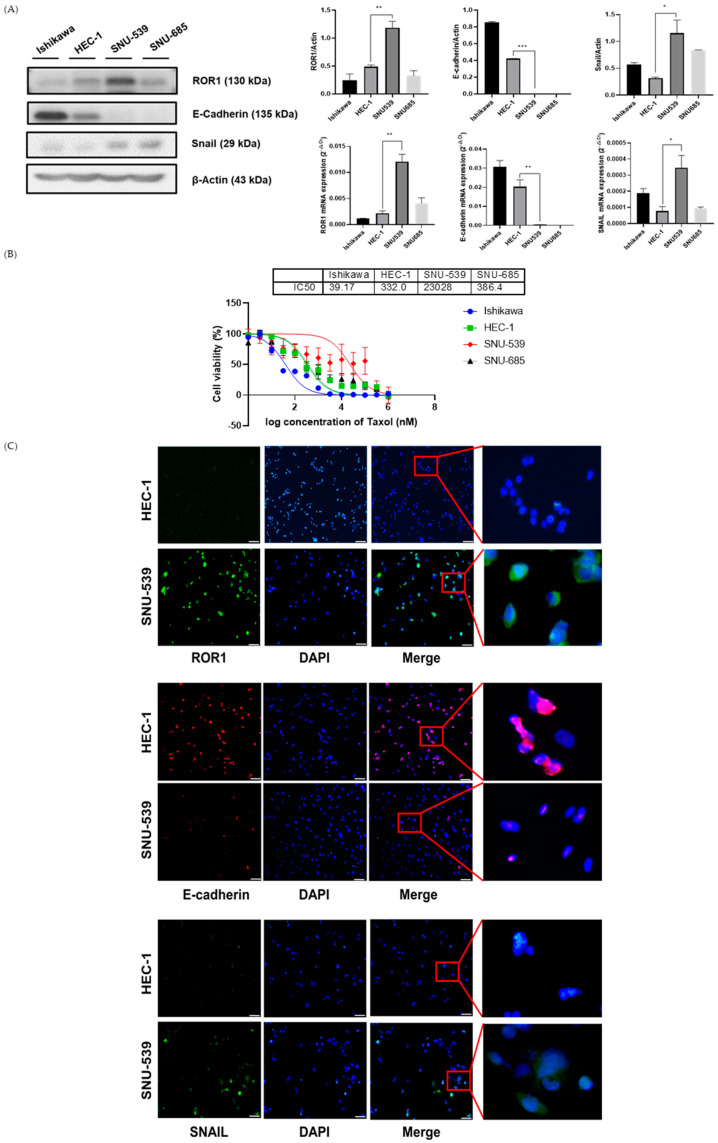
ROR1 and epithelial−mesenchymal transition (EMT) markers in endometrial cancer cells. (**A**) Expression of ROR1 and EMT markers; (**B**) Investigation of anticancer drug resistance in endometrial cancer cells; (**C**) The localization of ROR1 and EMT markers with immunofluorescence. (blue−DAPI, green−ROR1, Snail, red−E-cadherin) Error bars represented the mean ± SEM of at least three independent experiments. * *p* ≤ 0.01, ** *p* ≤ 0.001, *** *p* ≤ 0.0001.

**Figure 2 medicina-59-00994-f002:**
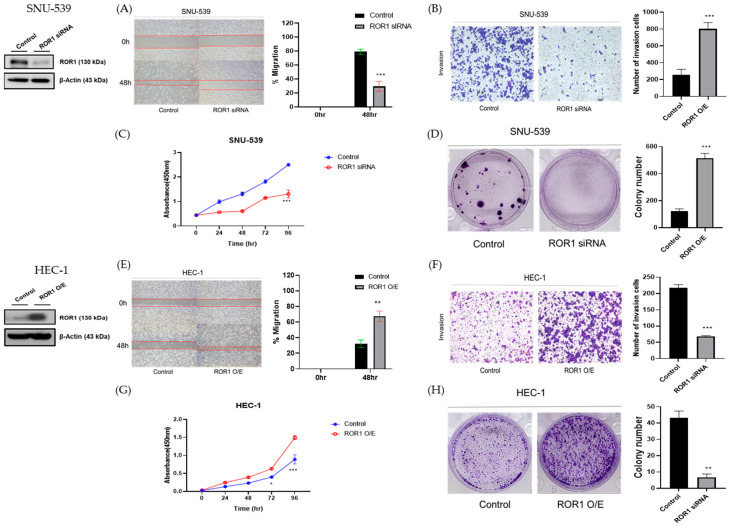
The transition of ROR1 expression contributes to metastasis and EMT formation in endometrial cancer cells. Silencing ROR1 with ROR1 siRNA was achieved in the SNU-539 cell line, and lower ROR1 expression was shown to be related to less cell migration and proliferation. This was achieved using a migration assay (**A**), invasion assay (**B**), cell proliferation assay (**C**), and colony assay (**D**). ROR1 overexpression was achieved in the HEC-1 cell line and showed that higher ROR1 expression is related to greater cell migration and proliferation (**E**–**H**). Error bars represented the mean ± SEM of at least three independent experiments. * *p* ≤ 0.05, ** *p* ≤ 0.001, *** *p* ≤ 0.0001.

**Figure 3 medicina-59-00994-f003:**
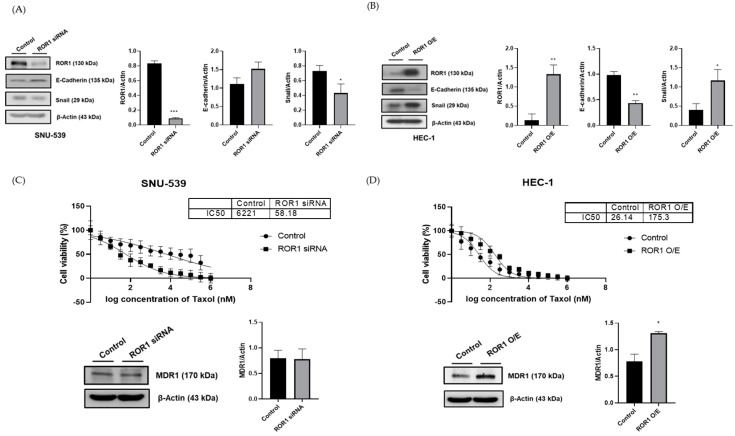
EMT markers expression and drug resistance in SNU−539 and HEC−1 cell lines. In the SNU−539 cell line, ROR1 silencing was related to less EMT formation (**A**) and less drug resistance (**B**). ROR1 overexpression in the HEC−1 cell line was related to greater EMT formation (**C**) and greater drug resistance (**D**). Error bars represented the mean ± SEM of at least three independent experiments. * *p* ≤ 0.05, ** *p* ≤ 0.01, *** *p* ≤ 0.0001.

**Figure 4 medicina-59-00994-f004:**
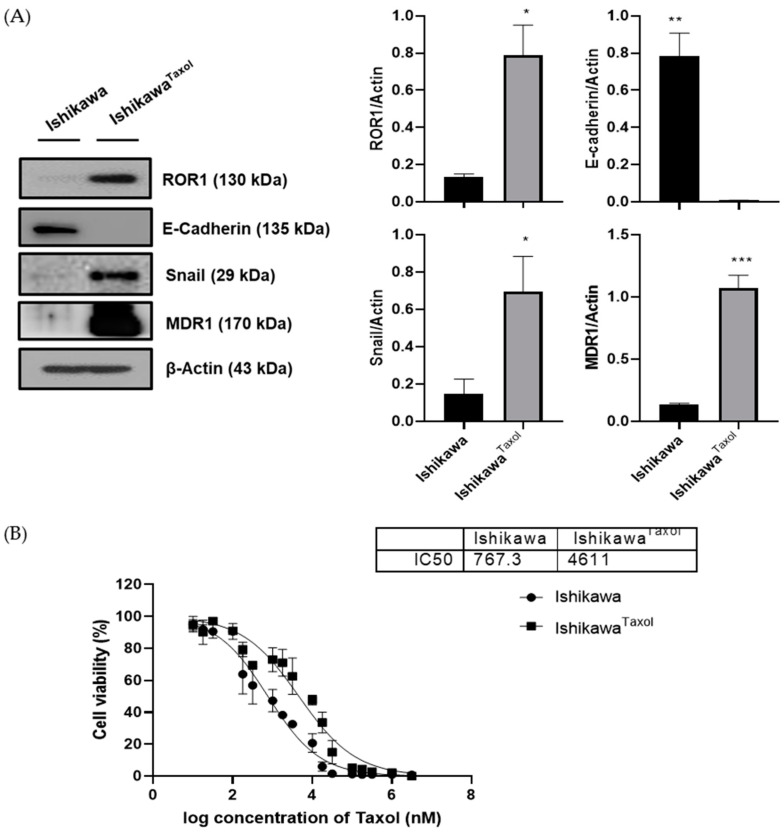
The expression of ROR1 and EMT markers in Ishikawa and Ishikawa^Taxol^ cell lines and the investigation of anticancer drug resistance. Higher ROR1 expression and EMT formation were identified in the Ishikawa^Taxol^ cell line, which has resistance to paclitaxel (**A**). Additionally, the IC50 of the Ishikawa^Taxol^ cell line was higher than that of the Ishikawa cell line (**B**). Error bars represented the mean ± SEM of at least three independent experiments. * *p* ≤ 0.01, ** *p* ≤ 0.001, *** *p* ≤ 0.0001.

**Table 1 medicina-59-00994-t001:** Experimental values of ROR1 and EMT markers expression in SNU-539 and HEC-1 cell line.

	Mean Value in SNU-539	Mean Value in HEC-1	Difference between Means	SEM ^1^	*p*-Value
ROR1	1.184	0.486	0.698	0.06964	0.0006
E-cadherin	0	0.4214	−0.4214	0.0008206	<0.0001
Snail	1.154	0.3185	0.8355	0.1406	0.004
ROR1 mRNA	0.01205	0.002155	0.009893	0.0008706	0.0003
E-cadherin mRNA	0.0003227	0.02025	−0.01993	0.002096	0.0007
Snail mRNA	0.0004787	0.0001446	0.0003341	0.0003347	0.0015

^1^ SEM, Standard Error of the Mean.

**Table 2 medicina-59-00994-t002:** Experimental values of ROR1, EMT markers, and MDR1 expression in SNU-539 cell line.

	Control Group	ROR1-Silenced Group	Difference between Means	SEM ^1^	*p*-Value
ROR1	0.8364	0.09228	−0.7441	0.01784	<0.0001
E-cadherin	1.113	1.522	0.4089	0.2478	0.1498
Snail	0.7329	0.4329	−0.3001	0.08331	0.0227
MDR1	0.7977	0.7751	−0.02267	0.1486	0.8862

^1^ SEM, Standard Error of the Mean.

**Table 3 medicina-59-00994-t003:** Experimental values of ROR1, EMT markers, and MDR1 expression in HEC-1 cell line.

	Control Group	ROR1-Silenced Group	Difference between Means	SEM ^1^	*p*-Value
ROR1	0.1344	1.333	1.199	0.1681	0.002
E-cadherin	0.9837	0.4346	−0.5491	0.08161	0.0025
Snail	0.4014	1.165	0.7639	0.1919	0.0164
MDR1	0.7778	1.312	0.5340	0.1425	0.02

^1^ SEM, Standard Error of the Mean.

**Table 4 medicina-59-00994-t004:** Experimental values of ROR1, EMT markers, and MDR1 expression in Ishikawa and Ishikawa^Taxol^ cell lines.

	Ishikawa	Ishikawa^Taxol^	Difference between Means	SEM ^1^	*p*-Value
ROR1	0.1318	0.7896	0.6578	0.09331	0.0021
E-cadherin	0.7972	0	−0.7972	0.06707	0.0003
Snail	0.1946	0.7748	0.5802	0.06863	0.0011
MDR1	0.1333	1.032	0.8987	0.03798	<0.0001

^1^ SEM, Standard Error of the Mean.

## Data Availability

Not applicable.

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
