# Peer review of "The Role of ROR1 in Chemoresistance and EMT in Endometrial Cancer Cells"

_medicina, 2023, doi:10.3390/medicina59050994_

Round 1

Reviewer 1 Report (Previous Reviewer 3)

Authors have made necessary corrections to the manuscript but some issues still remain.

The results presented in Supplementary Excel Table need to be presented in a single Table in Word and not eight-sheet Excel file. Moreover, this Table should be an integral part of the manuscript and not just the supplementary material as the presented data is important.  

Final spelling and grammar check is suggested before publication.

Author Response

Thank you for your kindly comment. As your comments, the experimental values were presented in the manuscript as a ‘Table’ or content., not as supplementaries. The revised contents were highlighted with yellow. Thank you for your comment.

Reviewer 2 Report (Previous Reviewer 1)

Thank you for inviting me to review the manuscript "The role of ROR1 in chemoresistance and EMT in endometrial 2 cancer cells" by Kyung-jun Lee et al.

The authors aim to investigate the role of Receptor tyrosine kinase-like orphan receptor type 1 (ROR1) in chemoresistance in endometrial cancer cell lines.

The abstract is concise and summarises the content of the article well. A good revision of the written English has been carried out. The introduction has been improved and the molecular classification of endometrial cancer has been included. References are up to date. Figures are explanatory 

Although the usefulness of this line of research is limited in this type of cancer, the article has been improved enough to be interesting.

Author Response

Thank you for your kindly comment. As your comment,  the recent reference were included in the manuscript and introduction has been reinforced more firmly. 

Thank you again.

This manuscript is a resubmission of an earlier submission. The following is a list of the peer review reports and author responses from that submission.

Round 1

Reviewer 1 Report

Thank you for inviting me to review the manuscript "The role of ROR1 in chemoresistance and EMT in endometrial 2 cancer cells" by Kyung-jun Lee et al.

The authors aim to investigate the role of Receptor tyrosine kinase-like orphan receptor type 1 (ROR1) in chemoresistance in endometrial cancer cell lines.

The abstract is concise and summarises the content of the article well. The English is well written and only minor linguistic revision is required. References are up to date. Figures are explanatory.

The article is confusing to read. The content is not novel. The role of these receptors in the chemoresistance of other tumours is well known. Also chemotherapy is not the main way of treatment of endometrial cancer and surgery and radiotherapy are more relevant.

Reviewer 2 Report

Dear Authors,

thank you for submitting this interesting manuscript to Medicina. Your research presents novel approach to EC.

Unfortunately there is quite a lot language mistakes which make it difficult to read. Please improve it carefully.

In the introduction when you explain the histopatological divission of EC you fail to mention current approach described in:

https://www.google.com/url?sa=t&rct=j&q=&esrc=s&source=web&cd=&ved=2ahUKEwjn-pTqoo7-AhXQnIsKHTZtAOMQFnoECBEQAQ&url=https%3A%2F%2Fijgc.bmj.com%2Fcontent%2Fijgc%2F31%2F1%2F12.full.pdf&usg=AOvVaw05wzU3f1fGz3FRjYeBA8Ud

Please include it in your deliberations.

Reviewer 3 Report

The aim of this study was to investigate the role of ROR1 through in vitro experiments in endometrial cancer cell lines. This is a very interesting and innovative research. Still, some issues should be addressed before publication.

Authors stated that obtained results are presented as mean +/- SD, but it cannot be seen clearly in the manuscript. Authors should clearly present values obtained in the experiments.

In the results section p values of group comparisons should be presented in order to confirm the findings and statements made by the authors.

If authors claim that relationships between tested parameters were found than they should present the findings of preformed correlations as well. Moreover, if correlation was performed it should be stated in the paragraph regarding statistical analysis. Otherwise, authors should only discus the differences between groups and not associations.  

Moreover, presenting obtained results in a table is suggested for better clarity.

English editing by a native or professional speaker is suggested.